

**Structure and evolution of the drainage system of a Himalayan debris-**
**covered glacier, and its relationship with patterns of mass loss**
Douglas I. Benn[1], Sarah Thompson[2], Jason Gulley[3], Jordan Mertes[4], Adrian Luckman[2] and
Lindsey Nicholson[5]
[1] *School of Geography and Sustainable Development, University of St Andrews, UK*
[2] *Department of Geography, Swansea University, Swansea, UK*
[3] *School of Geosciences, University of South Florida, FL, USA*
[4] *Department of Geological and Mining Engineering and Sciences, Michigan Tech, MI, USA*
[5] *Institute for Atmospheric and Cryospheric Sciences, University of Innsbruck, Austria*
**Abstract**
This paper provides the first synoptic view of the drainage system of a Himalayan debris-
covered glacier and its evolution through time, based on speleological exploration and
satellite image analysis of Ngozumpa Glacier, Nepal. The drainage system has several linked
components: 1) a seasonal subglacial drainage system below the upper ablation zone; 2)
supraglacial channels allowing efficient meltwater transport across parts of the upper ablation
zone; 3) sub-marginal channels, allowing long-distance transport of meltwater; 4) perched
lakes, which intermittently store meltwater prior to evacuation via the englacial drainage
system; 5) englacial cut-and-closure conduits, which may undergo repeated cycles of
abandonment and reactivation; 6) a 'base-level' lake system (Spillway Lake) dammed behind
the terminal moraine. The distribution and relative importance of these elements has evolved
through time, in response to sustained negative mass balance. The area occupied by perched
lakes has expanded upglacier at the expense of supraglacial channels, and Spillway Lake has
grown as more of the glacier surface ablates to base level. Subsurface processes play a
governing role in creating, maintaining and shutting down exposures of ice at the glacier





surface, with a major impact on spatial patterns and rates of surface mass loss. Comparison of
our results with observations on other glaciers indicate that englacial drainage systems play a
key role in the response of debris-covered glaciers to sustained periods of negative mass
balance.
**1. Introduction**
Debris-covered glaciers in many parts of the Himalaya have undergone significant surface
lowering in recent decades, with net losses of several tens of metres since the 1970s (Bolch et
al., 2008a, 2011; Kääb et al., 2012). Glacier thinning and reduced surface gradients have
resulted in lower driving stresses and ice velocities, and large parts of many glaciers are now
stagnant or nearly so (Bolch et al., 2008b; Quincey et al., 2009). These morphological and
dynamic changes have encouraged formation of supraglacial lakes and increased water
storage within glacial hydrological systems (Reynolds, 2000; Quincey et al., 2007; Benn et
al., 2012). Where lakes form behind dams of moraine and ice, volumes of stored water can be
as high as $10^8$ m$^3$, in some cases posing considerable risk of glacier lake outburst floods
(GLOFs) (Yamada, 1998; Richardson and Reynolds, 2000; Kattelmann, 2003).
Several studies have shown that the development and enlargement of englacial conduits play
an important role in the evolution of debris-covered glaciers during periods of negative mass
balance (e.g. Clayton, 1964; Kirkbride, 1993; Krüger, 1994; Benn et al., 2001, 2009, 2012;
Gulley and Benn, 2007; Thompson et al., 2016). The collapse of conduit roofs can expose
areas of bare ice at the glacier surface, locally increasing ablation rates. Additionally, areas of
subsidence associated with englacial conduits create closed hollows (dolines) that can evolve
into supraglacial ponds and lakes, further increasing ice losses by calving. Conversely,
supraglacial lakes can drain if a connection is made with the englacial drainage system,
provided the lake is elevated above hydrological base level ('perched lakes'; Benn et al.,



2001). Drainage of relatively warm lake waters through the glacier leads to conduit
enlargement, which in turn increases the likelihood of roof collapse, surface subsidence and
ultimately new lake formation (Sakai et al., 2000; Miles et al., 2015). Because ablation rates
around supraglacial lake margins are typically one or two orders of magnitude higher than
that under continuous surface debris, lakes contribute disproportionately to overall rates of
glacier ablation (Sakai et al., 1998, 2000, 2009; Thompson et al., 2016). By controlling the
location and frequency of surface subsidence and lake drainage events, englacial conduits
strongly influence overall ablation rates, and the volume of water that can be stored in and on
the glacier (Benn et al., 2012).

Speleological investigations in debris-covered glaciers in the Khumbu Himal have
demonstrated that englacial conduits can form by three processes: 1) 'cut and closure' or the
incision of supraglacial stream beds followed by roof closure; 2) hydrologically assisted
crevasse propagation, or hydrofracturing, which may route water to glacier beds; and 3)
exploitation of secondary permeability in the ice (Gulley et al., 2009a, b; Benn et al., 2012).
The relative importance of these processes in the development of glacial drainage systems,
however, has not been investigated in detail. Furthermore, there are no data on the large-scale
structure of englacial and subglacial glacial drainage systems in the Himalaya, or how they
evolve during periods of negative mass balance. In this paper, we investigate the origin,
configuration and evolution of the drainage system of Ngozumpa Glacier, using three
complementary methods. First, speleological surveys of englacial conduits are used to
provide a detailed understanding of their formation and evolution. Second, historical satellite
imagery and high-resolution digital elevation models (DEMs) are used to identify past and
present drainage pathways, glacier-wide patterns of surface water storage and release, and
regions of subsidence. Finally, feature tracking on TerraSAR-X imagery is used to detect
regions of the glacier subject to seasonal velocity fluctuations, as a proxy for variations in



subglacial water storage. Taken together, these methods provide the first synoptic view of the
drainage system of a large Himalayan debris-covered glacier, and its influence on glacier
response to recent warming.

**2. Study area and methods**
Ngozumpa Glacier is located in the upper Dudh Kosi catchment, Khumbu Himal, Nepal (Fig.
1). It has three confluent branches: a western (W) branch flowing from the flanks of Cho Oyu
(8188 m); a north-eastern (NE) branch originating below Gyachung Kang (7922 m); and an
eastern (E) branch (Gaunara Glacier) nourished below a cirque of 6000 m peaks. The NE and
E branches are no longer dynamically connected to the main trunk, which is fed solely by the
W branch (Thompson et al., 2016). The equilibrium line altitude (ELA) is not well known.
Google Earth images from 3 November 2009 (after the end of the ablation season) and 9 June
2010 (at the beginning of the monsoon accumulation season) show bare ice up to ~5700 m
above sea level (a.s.l.) on all three branches, and this value is adopted as an approximate
value of the ELA.

The lower ablation zone of the glacier is stagnant, with little or no detectable motion on most
of the E branch, or on the main trunk for ~7 km upglacier of the terminus (Bolch et al.,
2008b; Quincey et al., 2009; Thompson et al., 2016). The lowermost 15 km of the glacier
(below ~5250 m a.s.l.) is almost completely mantled with supraglacial debris. The debris
cover thickens downglacier, reaching 1.80 ± 1.21 m near the terminus (Nicholson, 2004;
Nicholson and Benn, 2012). In common with other large debris-covered glaciers in the
region, Ngozumpa Glacier has undergone significant surface lowering in recent decades, and
the glacier surface now lies >100 m below the crestlines of the late Holocene lateral moraines
(Bolch et al., 2008a, 2011).





The lower tongue of the glacier has a highly irregular surface, with numerous closed basins
separated by mounds, ridges and plateaux with a relative relief of ~50 m (Fig. 2). Most basins
contain supraglacial ponds and lakes, which typically persist for a few years before draining
(Benn et al., 2001; 2009, 2012; Gulley and Benn, 2007). Near the terminus of Ngozumpa
Glacier, a system of lakes is ponded behind the terminal moraine (informally named Spillway
Lake; Fig. 1). This lake system increased in area by around 10% per year from the early
1990s until 2009, but between 2009 and 2015 experienced a reduction of area and volume as
a result of lake level lowering and redistribution of sediment (Thompson et al., 2012, 2016;
Mertes et al., 2016). This hiatus is likely to be temporary and continued growth of the lake is
expected in the coming years, as has been the case with other 'base-level lakes' in the region
(Sakai et al., 2009).

We surveyed 2.3 km of englacial passages in Ngozumpa Glacier, using standard
speleological techniques modified for glacier caves (Gulley and Benn, 2007). Conduit
entrances were identified during systematic traverses of the glacier surfaces. Within each
conduit, networks of survey lines were established by measuring the distance, azimuth and
inclination between successive marked stations using a Leica Distomat laser rangefinder and
a Brunton Sightmaster compass and inclinometer. Scaled drawings of passages in plan,
profile and cross-section were then rendered *in situ*, and include observations of
glaciostructural and stratigraphic features exposed in passage walls, thereby allowing the
origin and evolution of conduits to be reconstructed in detail. In this paper, we focus on five
conduits, which exemplify different stages of conduit formation, abandonment and
reactivation. Three of the conduits have been previously described by Gulley and Benn
(2007), but in this paper we revise our interpretation of their origin in some important
respects. Some of the conduits drained water from or fed water into supraglacial lakes, and in



some cases it was possible to relate phases of conduit development to specific lake filling or
drainage events, identified in satellite images.

A range of optical imagery was used to map indicators of the large-scale structure of the
drainage system (Table 1). The location of supraglacial channels and ephemeral supraglacial
ponds were mapped using declassified Corona KH-4 imagery from 1965, Landsat 5 TM
(2009), GeoEye-1 (9 June 2010 and 23 December 2012) and WorldView-3 (5 January 2015)
imagery. The Corona and Landsat imagery was not co-registered or orthorectified beyond the
standard terrain correction of the product, and was used to identify the presence / absence of
larger lakes or channels and not to quantify rates of change.

Geo-Eye-1 imagery from June 2010 and December 2012, and Worldview-3 imagery from
January 2015 were acquired for a region covering 17.4 km$^2$ of the ablation area of the glacier.
Three stereoscopic DEMs of 1 m resolution were constructed from the stereo multispectral
imagery using the PCI Geomatica Software Package, and used to determine spatial patterns
of elevation change. The construction and correction of the DEMs is discussed in detail in
Thompson et al. (2016).

The 2010 DEM was used to define the extent of individual surface drainage basins on the
glacier surface. This was achieved by identifying surface elevation contours that entirely
surround other contours of a lesser height. Each supraglacial catchment was then defined by
the crestlines of ridges that separate the closed basins. Initially, we used 2 m contours but
these produced a large number of very small 'basins', due to the high roughness of the
bouldery glacier surface. Subsequently, we used 5 m contours that yielded a set of closed
basins that closely matched the location of ephemeral supraglacial ponds and lakes on the



glacier surface. The extent of many basins changed between 2010 and 2015 due to ice-cliff
backwasting, although all basins persisted through the period covered by the DEMs.

Glacier surface velocities were derived using feature tracking between synthetic aperture
radar images acquired by the TerraSAR-X satellite on 19 September 2014, 18 and 29 January
2015 and 5 January 2016. Feature tracking was done using the method of Luckman et al.
(2007), which searches for a maximum correlation between evenly spaced subsets (patches)
of each image giving the displacement of glacier surface features which are converted to
speed using time delay between images. Image patches were ~400 m x 400 m in size and
sampled every 40 m producing a spatial resolution of between 40 and 400 m depending on
feature density. Based on feature matching precise to one pixel (2 m), precision of the
measured velocities is 0.006 m day$^{-1}$ over the annual (341 day) period and 0.018 m day$^{-1}$ over
the winter (111 day) period.

**3. Mechanisms of englacial conduit formation**
To provide a comprehensive view of processes of englacial conduit formation on the glacier,
we describe two sites in detail (NG-04 and NG-05), then briefly describe and reinterpret three
previously published sites (NG-01, NG-02 and NG-03; Gulley and Benn, 2007).

*3.1 NG-04*
*Description:* Conduit NG-04 (27°57'24"N, 86°41'55" E; 4805 m a.s.l.) was surveyed in
November 2009, and consisted of a main passage (A; Fig. 3) and two shorter side-passages
(B and C) leading off to the west. The main passage extended from a large hollow on the
glacier surface (Basin C-63 on Fig. 9a) for a distance of at least 473 m (Fig. 2e), where the
survey was discontinued due to deep standing water on the cave floor. Side-passage B also
connected with a basin on the glacier surface (Basin W-6, Fig. 9b). Side-passage C was at



least 25 m long, but was not surveyed beyond this distance due to the evident instability of
the highly fractured walls.

The main passage consisted of an upper level with a flat or gently inclined floor, and a lower
narrow incised canyon. The passage was highly sinuous, with a sinuosity in the surveyed
reach of 5.52. Near A4 (Fig. 3), there was a tight cutoff meander loop off the main passage
(Fig. 4a). The base of the abandoned loop had a flat floor and lacked the incised lower level
that was present elsewhere in the system. The upper floor level could also be traced along the
walls of side passages B and C, which we interpret as twin remnants of a second meander
cutoff. The floor of the upper level sloped gently downward from A1 to A14, rose from there
to between A18 and A19, after which it descended once more. Sandy bedforms on the floor
and scallops on the ice walls of this upper level indicate that water flow was from A1 towards
A21.

Passage morphology in the upper level was very variable, including tubular, box-shaped,
triangular and irregular sections (Figs. 3 & 4b-d). Throughout most of the system, planar
structures were visible in the ceiling or walls of the upper level, running parallel to the
passage axis with variable inclination. The structures took the form of: (1) 'sutures' at the
line of contact between opposing walls (S: Fig. 3; Fig. 4b, c), (2) intermittent narrow voids
(V: Fig. 3; Fig. 4c), and (3) bands of sorted sand or gravel a few cm thick (SB: Fig. 3; Fig.
4d). Some of the voids increased in width inward, in some cases opening out into gaps tens of
cm across. In some places, bands of sorted sediment could be traced laterally into open voids
or sutures. At several points along the main passage, a pair of planar structures occurred on
opposite walls of the passage. Side-passage B had a narrow, meandering seam of dirty ice
running along its ceiling, and in Passage C the walls tapered upward to meet at a ceiling
suture.





The floor of the incised lower level in both parts of the main passage sloped down towards
side passages B and C (arrows, Fig. 3). A pair of incised channels was confluent at C1,
whereas a single incised channel was present in passage B, where its lower (western) end was
blocked by an accumulation of ice and debris.
*Interpretation:* The partially debris-filled structures in the walls and ceiling of the upper level
are closely similar to many examples of canyon sutures we have observed in cut-and-closure
conduits in the Himalaya and Svalbard, marking the planes of closure where former passage
walls have been brought together by ice creep and/or blocked by ice and debris (cf. Gulley et
al., 2009a, b). Cut-and-closure conduits are typically highly sinuous and have variable cross
sectional morphologies, ranging from simple *plugged canyons* (incised channels with roofs of
névé), to *sutured canyons* (partially or completely closed by ice creep), *horizontal slots*
(formed by lateral channel migration followed by roof closure), and *tubular passages* (where
passage re-enlargement has occurred under pipe-full (phreatic) conditions (Gulley et al.,
2009b). The tubular morphology of the upper passage in NG-04, combined with the sutures,
voids and sediment bands in the walls and ceiling indicates that the passage has been re-
enlarged under pipe-full conditions following an episode of almost complete closure. For
example, the sub-horizontal bands of sorted sand on both conduit walls between A15 and
A18 (Fig. 3d) suggest complete suturing of a low, wide reach (horizontal slot) prior to
formation of the surveyed passage.
The tubular and box-shaped cross-profiles and undulating long-profile of the upper passage
are consistent with fluvial erosion under pipe-full or phreatic conditions (cf. Gulley et al.,
2009b). This contrasts with the canyon-like form and consistent down-flow slope of channels
incised under atmospheric (vadose) conditions, typical of simple cut-and-closure conduits.

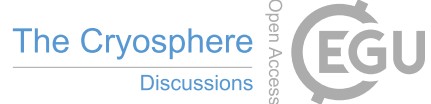

The dimensions of the upper passage (typically 2 m high and 3 m wide) are consistent with
high discharges. We conclude that the upper passage formed when water draining from a
supraglacial lake in Basin C-63 exploited the remnants of an abandoned cut-and-closure
conduit (Fig. 2e).

Following formation of the upper passage, the lower level was incised under vadose (non
pipe-full) conditions when the system accessed a new local base level via side-passages B
and C. We infer that this occurred when the cutoff meander loop between B1 and C2 was
exposed by ice-cliff retreat in Basin W-6. Water flow between A1 and B2 continued in the
same direction as before, but between A14 and A21 flow was reversed and discharge much
reduced.

Evolution of conduit NG-04 can be summarized as follows: 1) a cut-and-closure conduit was
formed by incision of a supraglacial stream; 2) this conduit was abandoned and almost
completely closed, presumably after it lost all or most of its source of recharge following
downwasting of the overlying glacier surface; 3) the conduit remnants were exploited and
enlarged by water draining from a supraglacial pond in Basin C-63; and 4) surface ablation in
Basin W-6 broke into the conduit, creating a new base level and initiating floor incision. This
remarkable cave illustrates how relict drainage systems can be reactivated when connected to
new sources of recharge, and demonstrates how patterns of drainage can change dramatically
within a single system in response to changing surface topography.

*3.2 NG-05*
*Description:* In December 2009 a conduit portal was exposed in an ice cliff at the margin of
Spillway Lake (27°56′36″ N, 86°42′46″ E, 4670 m a. s. l.). This portal was one of the two
main efflux points that carried water into the lake from upglacier (Thompson et al., 2016).



Access to the conduit could be gained via the frozen lake surface. However, the lake ice was
broken up each morning by debris falling from the melting glacier surface above, severely
limiting the time available for survey. Consequently, only a short section could be mapped
(Fig. 5). The conduit had two main levels, separated by a narrow, partially ice-filled canyon.
The floor of the lower part was at lake level, and that of the upper level was 4.8 m higher,
close to the summer monsoon level of the lake, as indicated by shorelines exposed around the
lake margins. Several notches on the passage walls recorded intermediate water levels. The
ice cliff above the upper level was obscured by a mass of icicles, but observations inside the
cave showed that the roof tapered up into a narrow, debris band or suture (Fig. 6a).

*Interpretation:* Although short, this passage is important for understanding the drainage
system of Ngozumpa Glacier. The debris band and suture in the roof indicates that, like NG-
04, the passage formed by a process of channel incision and roof closure. Additionally, the
passage is graded to the seasonally fluctuating surface of Spillway Lake. We therefore
conclude that the main drainage on the eastern side of the glacier consists of a cut-and-
closure conduit graded to the hydrologic base level of the glacier. For several km upglacier of
the portal, the debris-covered ice surface is highly irregular and broken into numerous closed
basins, implying that the conduit evolved from a surface stream that predates significant
downwasting of the glacier. The significance of these conclusions will be discussed later in
the paper.

*3.3 NG-01, 02 and 03*
*Description:* NG-01, NG-02 and NG-03 were mapped in December 2005, and described by
Gulley and Benn (2007) (Fig. 2e). NG-01 had carried water southward into a large basin on
the glacier surface (Basin C-25, Fig. 9a), whereas NG-02 drained water southward out of the
basin. NG-01 (27°57'58" N, 86°41'50" E) was a sinuous canyon passage with three main





levels. Debris bands cropped out in the walls of the uppermost level throughout its length,
either at the lateral margins of the passage or in the roof (Fig. 6b). The mid-level had a sub-
horizontal floor, into which the canyon linking to the lower level had been incised (Fig. 6c).
NG-02 (27°57'55" N, 86°41'51" E) was a sinuous canyon passage on two levels, extending in
a southwesterly direction from the basin. The upper level had a circular cross profile, and an
incised canyon beneath formed the lower level. A suture and debris band was exposed along
the entire length of the ceiling of the upper passage, mirroring the planform of the passage
(Fig. 6d). The lower level was an asymmetric flat-floored passage with a series of sills along
the margins. NG-03 (27°57'52" N, 86°42'02" E) consisted of a single passage graded to a
supraglacial pond in Basin E-19 (Fig. 2). Passage morphology changed from a low, wide
semi-elliptical cross-section to a more complex form with an elliptical upper section
separated by a narrow neck from a lower A-shaped section. At the top of the canyon, the
ceiling narrowed to a narrow slot, terminating in a band of coarse, unfrozen sandy debris.

*Interpretation:* For much of their length, all three conduits follow the trend of debris bands in
the walls or roof, leading Gulley and Benn (2007) to conclude that all were structurally
controlled. The debris bands were originally interpreted as debris-filled crevasse traces that
had been deformed during advection downglacier. When the original work was conducted,
the cut-and-closure model had not been developed, and we had yet to learn how to recognize
the diverse forms such conduits can take, especially in the later stages of their development.
It is now apparent that these conduits have all the hallmarks of cut and closure conduits. The
continuity and sinuous planform of the debris bands is consistent with formation by the
closure of incised canyons, rather than crevasse fills that had been deformed by ice flow.
Crevasses in the upper part of the glacier ablation area tend to be short, discontinuous and
oriented transverse to flow, unlike the observed debris bands in the conduit roofs, and ice



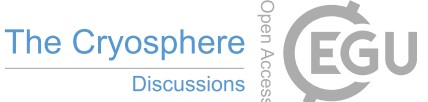

deformation is unlikely to be capable of generating the highly sinuous patterns observed
within the conduit debris bands.

We therefore reinterpret NG-01 – 03 as cut-and-closure conduits that have undergone cycles
of incision, abandonment, partial closure and later reactivation in response to fluctuating
patterns of recharge on the glacier surface. The circular and elliptical cross profiles observed
in NG-02 and NG-03 are consistent with phases of phreatic passage enlargement, analogous
to that in NG-04. Abandoned, incompletely closed conduits create hydraulically efficient
flow paths, which can be readily exploited and enlarged when surface ablation brings them
into contact with new sources of recharge.

**4. Drainage system structure**
In this section, we present evidence for the large-scale structure of the drainage system and
patterns of water storage and release, using X-band radar and optical satellite imagery and
high resolution DEMs from 2010, 2012 and 2015.

*4.1 Subglacial drainage system*
*Observations:* Direct observation of the subglacial drainage system was not possible. Instead,
we use seasonal fluctuations in glacier surface velocity to infer areas subject to variable
subglacial water storage. Mean daily ice velocities of the glacier between 29 January 2015
and 5 January 2016 are shown in Figure 7a. There is no detectable motion on the main trunk
within ~6 km of the terminus or on the lowermost 6 km of the E branch. The W branch is the
most active, with velocities of ~0.16 m day$^{-1}$ (~ 60 m yr$^{-1}$) at 5300 m a.s.l., declining to near
zero at 4900 m. The NE branch is slower, although velocities in its upper part could not be
determined due to image 'lay-over' in steep terrain. The active part of the NE branch does not
extend as far down as the confluence with the W branch, and a strip of stagnant ice ~100 -

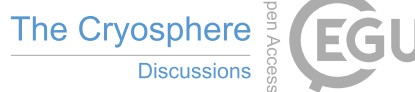



200 m wide extends ~3 km down the eastern side of the main trunk from the confluence
zone. Thus, neither the E nor the NE branch is dynamically connected to the main trunk.

Evidence for seasonal velocity fluctuations is shown in Fig. 7b, which shows mean daily
velocities between 29 January 2015 and 5 January 2016 (341 days) minus mean daily
velocities from 19 September 2014 to 18 January 2015 (111 days). Meteorological data from
the Pyramid Weather Station, at 5050 m a.s.l. c. 12 km east of Ngozumpa Glacier (available
through the Ev-K2-CNR SHARE program), indicate that air temperatures were consistently
below freezing between the 25th of September 2014 and the 28th of May 2015, defining a
minimum winter period for the upper ablation zone. The 111 day interval lies almost entirely
within the winter period but is less than half of its total duration, so Figure 7b yields
minimum values for a summer speed-up on the glacier. Most of the active parts of the glacier
exhibit some speed-up, although it is much more pronounced in some areas than others. On
the W branch, the greatest speed-up (by ~0.015 m day$^{-1}$ or ~10%) occurs above the
confluence with the NE branch. Areas of lesser speed-up also occur on the main trunk below
this point, although these are discontinuous and may be artifacts. Only the northern side of
the NE branch is affected by a seasonal speed-up. This area coincides with the tongue of
clean ice that descends through the icefall below Gyachung Kang (Fig. 1). Patchy areas of
apparent speed-up and slow-down occur elsewhere on the NE branch but may be artifacts. A
small speed-up also affects the active part of the E branch, above 5350 m a.sl.

*Interpretation:* The seasonal variations in ice velocities in the upper ablation zone are too
large to be explained by changes in ice creep rates, and are interpreted as variations in basal
motion (sliding and/or subglacial till deformation) in response to changing subglacial water
storage. This interpretation is supported by the spatial distribution of areas affected by the
seasonal speed-up, which coincide with, or occur downglacier of, heavily crevassed ice.





Much of the upper ablation area of Ngozumpa Glacier consists of icefalls with surface
gradients up to 30° (a, Fig. 8), and fields of transverse crevasses occur across the entire width
of the W branch down to an elevation of 5150 m (b). In contrast, crevasses are much less
widespread in the ablation zone of the NE branch, and occur only in the upper part of the
tongue of clean ice that descends from Gyachung Kang (c). Crevasses are almost absent on
the debris-covered part (d), which originates in two relatively low-altitude cirques. Fields of
transverse crevasses occur in the upper basin of the E branch, above ~5400 m. Crevasses
allow meltwater to be routed rapidly to the bed, and the existence of multiple recharge points
will encourage development of a distributed drainage system following the onset of the
monsoon ablation season. The lack of a clear seasonal velocity response on the lowermost 10
km of the glacier suggests that subglacial water is transported along the main trunk in
efficient conduits.

*4.2 Supraglacial channels*
*Observations:* Supraglacial stream networks are visible below the crevassed zones on all
three branches of the glacier. The most extensive network occurs on the tongue of clean ice
on the NE branch, where a set of sub-parallel channels descends from ~5180 m to the
junction with the W branch at ~4990 m (Fig. 2b, c). There are several discontinuous
supraglacial channels on the W branch between 5220 m and 5120 m a.s.l., including one
along the eastern margin of the glacier. Supraglacial channels occur on both flanks of the E
branch below ~5100 m a.s.l. The channels converge at the junction with the main trunk, and
after flowing over the glacier surface for several hundred metres the combined stream sinks
in a large hollow that is intermittently filled with water. Patterns of water storage in this
hollow are discussed in Section 4.4.



*Interpretation:* Perennial supraglacial channels can only persist if the annual amount of
channel incision exceeds the amount of surface lowering of the adjacent ice (Gulley et al.,
2009b). The rate at which ice-floored channels incise is controlled by viscous heat dissipation
associated with turbulent flow, and increases with discharge and surface slope (Fountain and
Walder, 1998; Jarosch and Gudmundsson, 2012). Because supraglacial stream discharge is a
function of surface melt rate and melt area, significant channel incision requires large
catchment areas. Therefore, incised surface channels tend to occur only where potential
catchments are not fragmented by crevasses or hummocky surface topography (Fig. 2). At
present, these conditions are met in relatively limited areas of Ngozumpa Glacier, below
crevassed areas and above hummocky debris-covered areas.

*4.3 Hummocky debris-covered areas and perched lakes*
*Observations:* Most of the lower ablation zone of the glacier (below ~5000 m) consists of
hummocky debris-covered topography. In this zone, the glacier surface is broken up into
numerous closed depressions, each of which forms a distinct drainage basin (Fig. 2d, e). Not
including the Spillway Lake basin that drains externally, in 2010 there were 111 surface
basins in the hummocky debris-covered zone (Fig. 9). The basins along the east and west
margins of the glacier form a series of depressions within almost continuous lateral troughs,
and are considered in Section 4.4. Here, we focus on the basins in the central part of the
glacier (C1 - C69; Fig. 9a) and the terminal zone (T1 - 6, Fig. 9b).

Of the 70 basins in the central part of the glacier, 56 (80%) contained ponds or lakes in at
least one of the three years covered by the Geo-Eye and Worldview imagery. Fifteen of the
42 lakes present in 2010 (36%) had disappeared by 2012 or 2015, whereas 14 basins that
were empty in 2010 contained lakes in one or more of the later years. Almost all of the
remainder underwent partial drainage and /or refilling. In contrast, the 5 lakes in the terminal

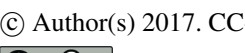



zone of the glacier (below Spillway Lake) exhibited great stability. Four showed no
significant change in area between 2010 and 2015, while the other showed an increase in
area.

*Interpretation:* Observations on and below the glacier surface show that drainage of perched
lakes occurs when part of the floor is brought into contact with permeable structures in the
ice (Benn et al., 2001; Gulley and Benn, 2007). The characteristics of NG-01 - 05 (which all
occur within the hummocky debris-covered zone) show that relict cut-and-closure conduits
are the dominant cause of secondary permeability in the glacier, providing pre-existing lines
of weakness along which perched lakes can drain.

The spatial extent and high temporal frequency of perched lake drainage events on the glacier
(Fig. 9a) imply a high density of relict conduits within the ice. A rough estimate can be
obtained by dividing the number of complete and partial drainage events (35) by the total
area of basins in the central part of the glacier (4.62 km$^2$), yielding ~7.6 relict conduit reaches
per km$^2$. This is a minimum estimate, because additional conduit remnants could occur below
and beyond the margins of observed lakes. Conversely, the number of lake filling events (23
over the 5 ablation seasons spanned by the imagery) shows that drainage routes commonly
become blocked. Conduit blockage processes have been described by Gulley et al. (2009b),
and include accumulation of icicles or floor-ice at the end of the melt season and creep
closure of opposing conduit walls. The interplay between drainage events and conduit
blockage maintains a dynamic population of supraglacial lakes, which contribute
significantly to ablation of the glacier, through absorption of solar radiation and ice melt, and
calving (Thompson et al., 2016).

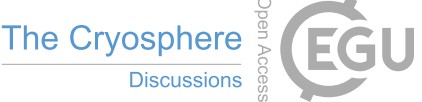

The stability of lakes in the terminal zone probably reflects a combination of factors. These
lakes are flanked by stable slopes of thick debris, which inhibit lake growth by melt or
calving. Furthermore, the lakes are located at or close to the hydrologic base level of the
glacier, determined by the terminal moraine that encircles the glacier terminus, inhibiting
drainage via relict conduits.

*4.4 Sub-marginal drainage*
*Observations:* Elevation differences between successive DEMs indicate linear zones of
enhanced surface lowering along both margins of Ngozumpa Glacier, where troughs extend
along the base of the bounding lateral moraines (Thompson et al., 2016; Fig. 10). The inner
moraine slopes consist of unvegetated, unconsolidated till, and undergo active erosion by a
range of processes including rockfall, debris flow and rotational landslipping (Benn et al.,
2012; Thompson et al., 2016). Although the debris eroded from the moraine slopes is
transferred downslope into the troughs, the troughs underwent surface lowering of 6 – 9 m
from 2010 to 2015, with a total annual volume loss in the moraine-trough systems of 0.4 x
$10^6$ $m^3$ $yr^{-1}$ (Thompson et al., 2016). This implies that a large volume of ice, debris or both is
evacuated annually from below the lateral margins of the glacier.

The lateral troughs form a series of closed basins, 12 on the west side and 22 on the east (Fig.
9b). Eight of the basins in the west trough and 17 of those in the east contained a lake in
2010, 4 (W) and 7 (E) of which had completely drained by 2012 or 2015. Four new lakes
appeared in the eastern trough in 2012 or 2015, and 1 (W) and 7 (E) underwent partial
drainage and/or refilling. Three basins on western side and one on the eastern side showed no
fluctuations in lake area.





Benn et al. (2001) provided detailed descriptions of lake filling and drainage cycles in basins
W-7 and W-5 (Lakes 7092 and 7093 in their terminology). In October 1998, basin W-7
contained three shallow ponds, but by October 1999 the basin was occupied by a single large
lake and water level had risen by ~9 m. Lake area had increased from 17,890 m$^2$ to 52,550
m$^2$, with 36% of the increase attributable to backwasting and calving of the surrounding ice
cliffs. By September 2000, the lake had almost completely drained and only shallow ponds
remained. Lake drainage occurred via an englacial conduit, which had been exposed by
retreat of the lake margin.  A lake in basin W-5 also underwent fluctuations in area and depth
between 1998 and 2000, but did not completely drain during that time. Horodyskuj (2015)
used time-lapse photography and a pressure transducer to document rapid lake-level
fluctuations in this basin, including rises and falls of several metres within hours.

Short-term cycles of lake drainage and filling can also be demonstrated in other basins within
the lateral trough systems using optical satellite imagery. Figure 11 shows a series of images
of the east side of the glacier close to the junction with the E branch, where a supraglacial
stream (Section 4.2) flows into a closed depression in Basin E-11 (Fig. 9b). A pond
occupying the basin expanded in area between March and May 2009, but drained between
June and August. In 2015 there is little evidence of the pond in January but a large pond is
present in June.

*Interpretation:* Widespread, rapid subsidence along both margins of the glacier can be
explained by enlargement and episodic collapse of sub-marginal conduits (Thompson et al.,
2016). Potential internal ablation rates were calculated from energy losses associated with
runoff and supraglacial lake drainage, and the resulting value of 0.12 to 0.13 x 10$^6$ m$^3$ yr$^{-1}$ is
around 30% of the measured volume losses, the difference being at least partly attributable to
sediment evacuation by meltwater.






The sub-marginal conduits appear to be perennial features of the glacier drainage system.
Upwellings in Spillway Lake are active during the winter months, indicating that conduits
transport water routed via the glacier bed in addition to summer melt- and rainwater.
However, much of the lower ablation zone appears to be bypassed by the sub-marginal
conduits, as evidenced by widespread water storage in supraglacial lakes and ponds (Section
4.3). As noted above, water is intermittently discharged from lakes in the central part of the
glacier into the lateral troughs via englacial conduits.

Cycles of lake drainage and filling in lateral basins indicate intermittent connections between
surface catchments and the sub-marginal meltwater channels (Fig. 9b). In some cases,
drainage events can be directly attributed to exploitation of englacial conduits (Benn et al.,
2001). The hourly changes in lake level recorded by Horodyskuj (2015) cannot be explained
by conduit opening and blockage, and more likely reflect short-term fluctuations in recharge
from surface melt and water release from storage.

*4.5 Spillway Lake*
*Observations:* In 2010, the area of the Spillway Lake surface catchment was 0.8 km$^2$, of
which 0.27 km$^2$ was occupied by the lake system. All of the water leaving the glacier passes
through Spillway Lake, entering via portals or upwellings at or close to lake level, and
leaving via a gap in the western lateral moraine ~1 km from the glacier terminus (1: Fig. 12).
In 2009, conduit NG-05 (Fig. 5; Section 3.2) entered the NE corner of the lake and is
interpreted as the distal part of the eastern sub-marginal conduit. A second conduit portal
visible at the NW lake margin in the same year is interpreted as the efflux point of the
western sub-marginal stream. The evolution of the Spillway Lake system, and its

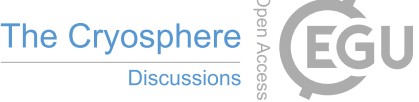

implications for drainage system structure in this part of the glacier, are examined in Section
5.4 below.

*4.6 Summary*
The evidence presented above demonstrates that the drainage system of Ngozumpa Glacier
comprises several linked elements: 1) the subglacial system; 2) supraglacial channels; 3) a
perched lake - englacial conduit system; 4) sub-marginal conduits; and 5) the Spillway Lake
system. These elements have a distinct spatial distribution (Fig. 13a). Evidence for seasonal
subglacial water storage is restricted to active parts of the glacier downglacier of crevasse
fields, where surface water can be routed to the bed. Supraglacial channels occur where
surface catchments and discharge are large enough to allow channel incision rates to outpace
surface ablation rates. Thus, perennial channels only occur where the glacier surface is not
broken up by crevasse fields or into small, closed basins. Perched lakes occur where the
glacier surface is broken up into closed basins, where the overall gradient of the glacier is
<3°. The life cycle of perched lakes is governed by the location of englacial 'cut and closure'
conduits and the frequency of connection and blockage events. Sub-marginal conduits occur
below both flanks of the glacier, and transport water from supraglacial channels, intermittent
drainage from perched lakes, and possibly the subglacial drainage system, into Spillway
Lake. The lake lies at the hydrologic base level of the glacier, and its extent reflects the
surface elevation of the glacier relative to the spillway through the terminal moraine.

**5. Evolution of the drainage system**
In this Section, we present evidence for changes in drainage system structure through time,
including features visible on Corona images from 1964 and 1965, speleological observations,
and repeat surveys of Spillway Lake since 1999.





*5.1 Supraglacial channels*
In 1964, a connected supraglacial drainage stream network was present on the eastern side of
the main trunk above the junction with the E branch (10 - 8 km from the terminus, 4950 m to
4920 m a.s.l.) (Fig. 14a). By 2010, this part of the glacier had been broken up into basins E-7,
E-8 and E-9, part of the lateral trough systems described in Section 4.4. Stream channels
were no longer present, although a number of isolated elongate ponds occur close to some of
the original channel locations (Fig. 14b). Sinuous depressions are visible in this area in the
DEMs from 2010, 2012 and 2015 (Fig. 14). The depressions have an overall reduction in
elevation to the south, but in detail they have up-and-down long profiles. In cross profile,
they are U-shaped and become wider and deeper through time (Fig. 14b).

We hypothesize that the supraglacial channels became deeply incised and transitioned into
cut-and-closure conduits, which continue to evacuate meltwater despite fragmentation of the
surface topography. Channel incision may have been encouraged by thickening debris cover
that would have reduced glacier surface lowering rates.

At the distal end of the eastern lateral trough, conduit NG-05 (Fig. 5) emerges into Spillway
Lake.  Passage morphology indicates that at this point the conduit formed by cut and closure
(Section 3.2).  Thus, there is evidence for a cut and closure origin of subsurface conduits at
both ends of the eastern lateral trough. We therefore infer that the sub-marginal conduits on
both sides of the glacier originated as supraglacial streams that became incised below the
surface. Such a scenario would require a continuous slope along both glacier margins. We
conclude that supraglacial streams occurred along both margins before development of the
current irregular topography, but transition to cut-and-closure conduits allowed these
drainage routes to persist after break-up of the glacier surface.



### 5.2 Englacial conduits in the hummocky debris-covered zone

Transition of drainages from supraglacial channels to cut-and-closure conduits appears to have been a widespread process on the glacier. The presence of sutures, planar voids and bands of sorted sediments in the ceilings and walls of conduits NG-01 - NG-05 indicate that they originated as supraglacial channels. As for the lateral channels, we infer that systems of supraglacial channels existed in the central part of the lower tongue before the glacier surface was broken up into small closed basins.

Differential surface ablation can eventually cause fragmentation and abandonment of cut-and-closure conduits, cutting off downstream reaches from former water sources. In abandoned reaches, processes of passage closure dominate over those of enlargement, and systems gradually shut down. Because cut and closure conduits are generally located close to the glacier surface, shut-down is commonly incomplete. Zones of narrow voids or sutures with infills of unfrozen sediment may persist, forming meandering lines of high permeability through otherwise impermeable glacier ice.

Reactivation will occur if a new water source becomes available, and a conduit remnant connects this source with a region of lower hydraulic potential. These conditions are met on stagnant, low-gradient glacier surfaces. Supraglacial lakes in closed basins provide both reservoirs of water and regions of elevated hydraulic potential. Drainage is highly episodic, and water may be stored in supraglacial lakes for years before passing farther down the system.

### 5.4 Spillway Lake

The recent history of Spillway Lake was discussed in detail by Thompson et al. (2012, 2016), and is briefly reviewed here. The present spillway through the SW side of the terminal





moraine has been in existence since at least 1965, when water emerged from the glacier and
entered a small pond behind the lateral moraine (1: Fig. 12). In the following decades, the
Spillway Lake system expanded upglacier from this point. On the Survey of Nepal map,
based on aerial photographs taken in 1992, the lake has a ribbon-like form, extending NE for
~600 m from the spillway. The lake had essentially the same outline at the time of our first
field survey in 1998, when water was observed to enter the lake via two subaerial portals and
an upwelling point (Fig. 12; Benn et al., 2001; Thompson et al., 2012). Between 1998 and
1999, several chasms and holes opened up on the glacier surface north of the western portal,
and by 2001 these had evolved into linear ponds and lakes (2: Fig. 12). Between 2001 and
2009, the Spillway Lake system underwent considerable expansion to the north, accompanied
by upglacier migration of the portal locations (3, 4: Fig. 12).

The predominantly linear patterns of lake expansion, and the location of meltwater portals
and upwellings, indicate that evolution of the Spillway Lake system was strongly
preconditioned by the locations of shallow englacial conduits (a and b: Fig. 12). Conduit
NG-05 (Section 3.4 and Fig. 5) and other examples exposed around the lake margins show
that the drainage system consists of cut-and-closure conduits graded to lake level. This near-
surface englacial conduit system provided pre-existing lines of weakness in the ice which,
when opened up to the surface by internal ablation and collapse, were exploited by ice-cliff
melting and calving processes.

Spillway Lake was thus established on a template provided by two englacial conduits (a & b,
Fig. 12), which were confluent prior to 1992. As it expanded upglacier, Spillway Lake
encroached on areas formerly occupied by perched lakes and incorporated former
supraglacial basins. A recent example is Basin C-33, which forms an inlier within the
Spillway Lake catchment (Figs. 9a and 12). This basin contained a perched lake in 2009 and



2010, but this drained prior to December 2012 and has not reformed. It is likely that this
basin will become entirely subsumed within the Spillway Lake catchment in the near future,
as a consequence of ice-cliff backwasting.

*5.5 Changing drainage patterns on the glacier*
Comparison of the drainage system structure in 2010 with evidence on Corona imagery from
1964 shows an upglacier expansion of the area occupied by closed depressions and perched
lakes, and the formation and upglacier expansion of the base-level Spillway Lake (Fig. 13b).
The widespread occurrence of cut-and-closure conduits (which originate by the incision of
surface streams) provides evidence of an even earlier stage in drainage evolution, when
supraglacial channels extended the full length of the glacier and closed basins were absent or
rare (Fig. 13c). Such a drainage system might have existed during the Little Ice Age, and
persisted into the early 20th Century.

Ngozumpa Glacier has thus responded to a prolonged period of negative mass balance with a
systematic reordering of its drainage system, characterized by less efficient evacuation of
meltwater and greater amounts of storage. More recent elements of the drainage system retain
a memory of older elements, and processes and patterns of ablation on the glacier continue to
be influenced by active and relict channels and conduits. Former supraglacial channels
preconditioned the location and density of cut-and-closure conduits, which in turn
precondition the formation and drainage of perched lakes and provide templates for the
expansion of Spillway Lake.

**6. Comparison with other glaciers**
Observations on other debris-covered glaciers in the Himalaya indicate that their drainage
systems share many of the characteristics described in this paper. Seasonal velocity





fluctuations have been documented on other large glaciers in the Mount Everest region and
on Lirung glacier, Nepal (Benn et al., 2012; Kraaijenbrink et al., 2016), indicating surface-to-
bed drainage and variations in subglacial water storage. Perennial supraglacial channels occur
in the upper ablation zones of many glaciers, in places where catchments are not fragmented
by crevasse fields or irregular surface topography (Gulley et al., 2009b; Benn et al., 2012).
Continuity between a supraglacial channel and an englacial cut-and-closure conduit has been
observed on Khumbu Glacier, clearly demonstrating the genetic relationship between the two
features (Gulley et al. 2009b). Perched lakes are widespread on Himalayan debris-covered
glaciers, and evidence for repeated filling and drainage (Watson et al., 2016; Miles et al.,
2017) suggest that englacial conduits may play an important role in their life cycles.
However, englacial conduits have only been explored in a few glaciers (Gulley and Benn,
2007; Gulley et al. 2009b; Benn et al. 2009), and much research remains to be done.
Similarly, very little is known about possible sub-marginal channels in Himalayan glaciers,
and our few attempts to enter these highly dynamic environments have been repulsed.

There is strong evidence on many glaciers that base-level lake growth is preconditioned by
englacial conduits. For example, upglacier expansion of the proglacial lake at Tasman
Glacier, New Zealand, has repeatedly echoed former chains of sink holes on the glacier
surface (Kirkbride, 1993; Quincey and Glasser, 2009). Recently formed chains of ponds on
the lower ablation zone of Khumbu Glacier, strongly suggests that the same process is
underway on that glacier (Watson et al., 2016). The integrated picture of drainage system
structure and evolution presented in this paper provides a framework for predicting what the
future has in store for Khumbu Glacier and other debris-covered glaciers in the region.

**7. Summary and Conclusions**





This paper has provided the first synoptic view of the drainage system of a Himalayan debris-covered glacier, including the spatial distribution of system components, their evolution through time, and their influence on processes and patterns of ablation. Our specific conclusions are as follows.

1) In the upper ablation zone, seasonal variations in ice velocity indicate routing of surface meltwater to the bed via crevasses, and fluctuations in subglacial water storage.

2) Systems of supraglacial channels occur where the glacier surface is uninterrupted by crevasses or closed depressions, allowing efficient evacuation of surface melt.

3) Active sub-marginal channels are evidenced by linear zones of subsidence along both margins of the glacier, and fluctuations in surface water storage and release. These channels likely formed from supraglacial channels by a process of cut and closure, and permit long-distance transport of meltwater through the ablation zone. Transport of sediment via the lateral channels destabilizes inner moraine flanks and delivers debris to the terminal zone, where it modulates ablation processes.

4) In the lower ablation zone (below ~5,000 m) the glacier surface consists of numerous closed drainage basins. Meltwater in this zone typically undergoes storage in perched lakes before being evacuated via the englacial drainage system. Englacial conduits in this zone evolved from supraglacial channels by a process of cut-and-closure, and may undergo repeated cycles of abandonment and reactivation.

5) Enlargement of englacial conduits removes ice mass that is not captured by surface observations until conduit collapse occurs, with the implication that observations of sudden surface lowering need not reflect sudden glacier mass loss over the same time period. Subsurface processes play a governing role in creating, maintaining and shutting down exposures of ice at the glacier surface, with a major impact on spatial patterns and rates of surface mass loss.



6) A large lake system (Spillway Lake) is dammed behind the terminal moraine, which forms
the hydrologic base level for the glacier. Since the early 1990s, Spillway Lake has expanded
upglacier, exploiting weaknesses formed by englacial conduits.
7) As part of the glacier response to the present ongoing period of negative mass balance, the
structure of the drainage system has changed through time, characterized by decreasing
efficiency and greater volumes of storage. Processes and patterns of ablation on the glacier
are strongly influenced by active and relict elements of the drainage system. Former
supraglacial channels evolved into cut-and-closure conduits, which in turn precondition the
formation and drainage of perched lakes and provide templates for the expansion of Spillway
Lake. Thus drainage elements that initially formed during earlier active phases of the glacier's
history continue to influence its evolution during stagnation.

**Acknowledgements**
Funding for ST was provided by the European Commission FP7-MC-IEF grant PIEF-GA-
2012-330805, and for LN by the Austrian Science Fund (FWF) Elise Richter Grant (V309).
Financial support for fieldwork in 2009 was provided by the University Centre in Svalbard
and a Royal Geographical Society fieldwork grant to ST. Field assistance was given by
Annelie Bergström and Alison Banwell. TerraSAR-X data were kindly provided by DLR
under Project HYD0178. The meteorological data were collected within the Ev-K2-CNR
SHARE Project, funded by contributions from the Italian National Research Council and the
Italian Ministry of Foreign Affairs, and we thank Patrick Wagnon of the IRD for collecting
and releasing the 2014-2015 data used in this paper.

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

No. 1, Data Centre for Glacier Research, Japanese Society of Snow and Ice, Japan, 96

pp.






Table 1: Satellite imagery used in the paper

| Sensor | Product type | Resolution m | Acquisition date | Cloud cover (%) |
|---|---|---|---|---|
| **Corona** | KH-4 | 3 | 04 Mar. 1965 | - |
| **Landsat 5 TM** | Level T1 | 30 | 05 Mar. 2009 | 17 |
| **Landsat 5 TM** | Level T1 | 30 | 08 May 2009 | 16 |
| **Landsat 5 TM** | Level T1 | 30 | 09 Jun. 2009 | 28 |
| **Landsat 5 TM** | Level T1 | 30 | 16 Aug. 2009 | 18 |
| **GeoEye-1** | GeoStereo<br>PAN/MSI | PAN 0.46<br>MSI 1.84 | 09 Jun. 2010 | 3 |
| **GeoEye-1** | GeoStereo<br>PAN/MSI | PAN 0.46<br>MSI 1.84 | 23 Dec. 2012 | 0 |
| **WorldView-3** | GeoStereo<br>PAN/MSI | PAN 0.46<br>MSI 1.84 | 05 Jan. 2015 | 0 |









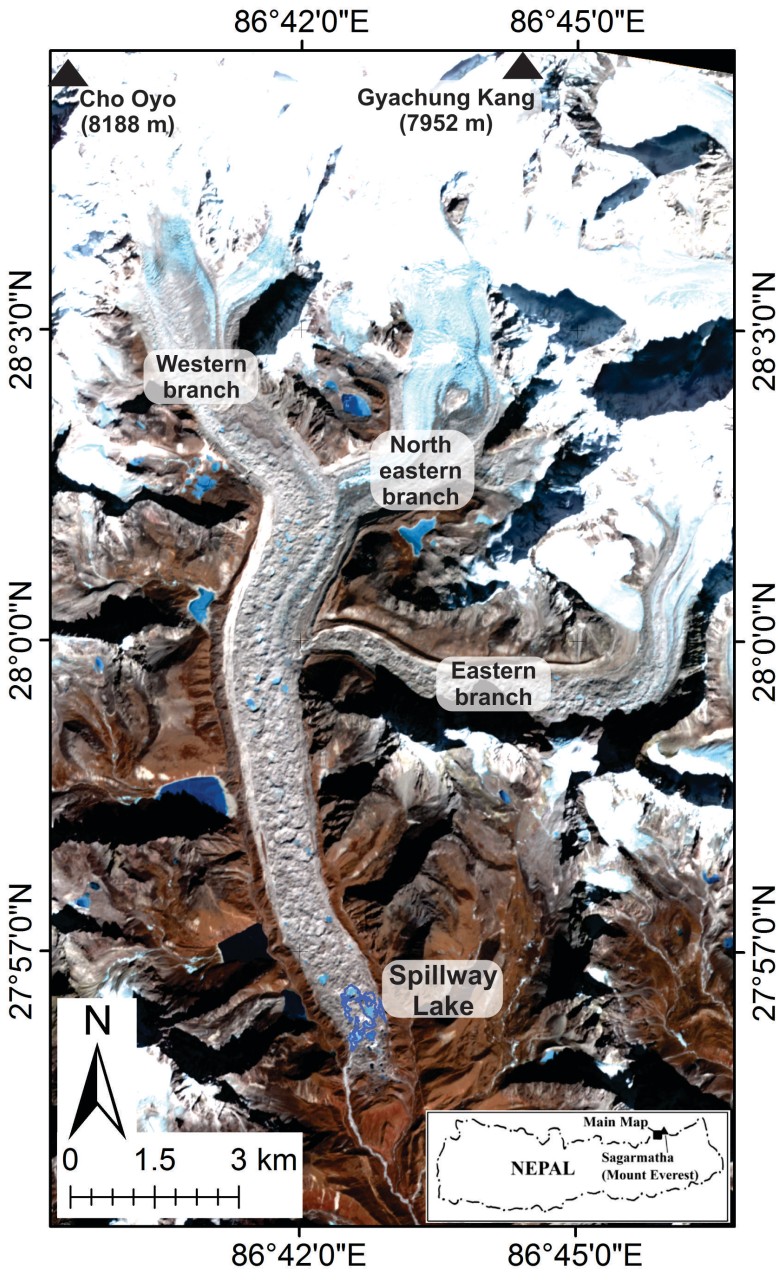

Fig. 1: Ngozumpa Glacier, showing the location of the three branches and Spillway Lake.
Image: orthorectified GeoEye-1 from December 2012.



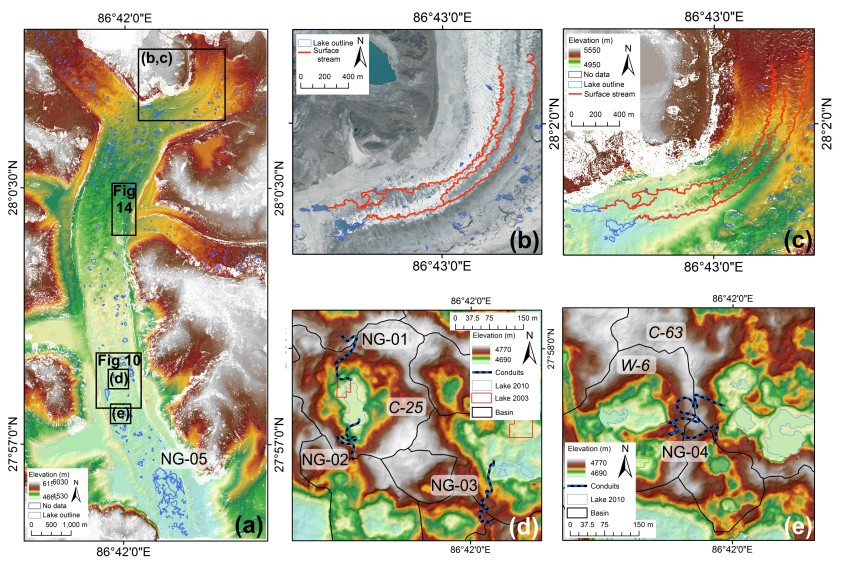

Fig. 2: Examples of surface topography, supraglacial meltwater channels and englacial conduit locations on Ngozumpa Glacier: a) DEM of the lower ablation zone of the glacier, showing location of panels b-e, Figs. 10 & 14, and englacial conduit NG-05; b) supraglacial channels shown on GeoEye-1 imagery from June 2010; c) the same area shown on the 2010 DEM; d) hummocky debris-covered ice showing the boundaries of closed surface basins and locations of englacial conduits NG-01 to NG-03. Considerable basin expansion occurred in the 4 ablation seasons between the conduit surveys (December 2005) and the date of the DEM (June 2010); e) hummocky debris-covered ice and location of englacial conduit NG-04 (surveyed November 2009, 7 months before the date of the DEM).

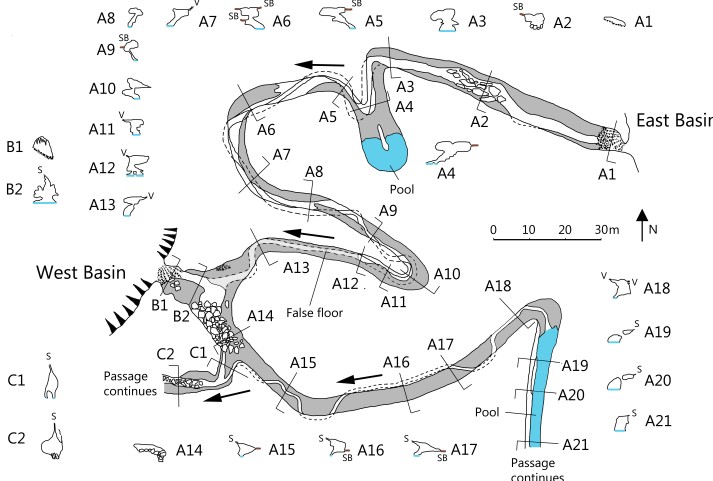



Fig. 3: Plan and passage cross sections, englacial conduit NG-04. SB: sediment band, S: suture, V: voids. Grey shaded areas indicate the floor of the upper level and blue areas indicate standing water. Arrows show the floor slope directions of the lower level.

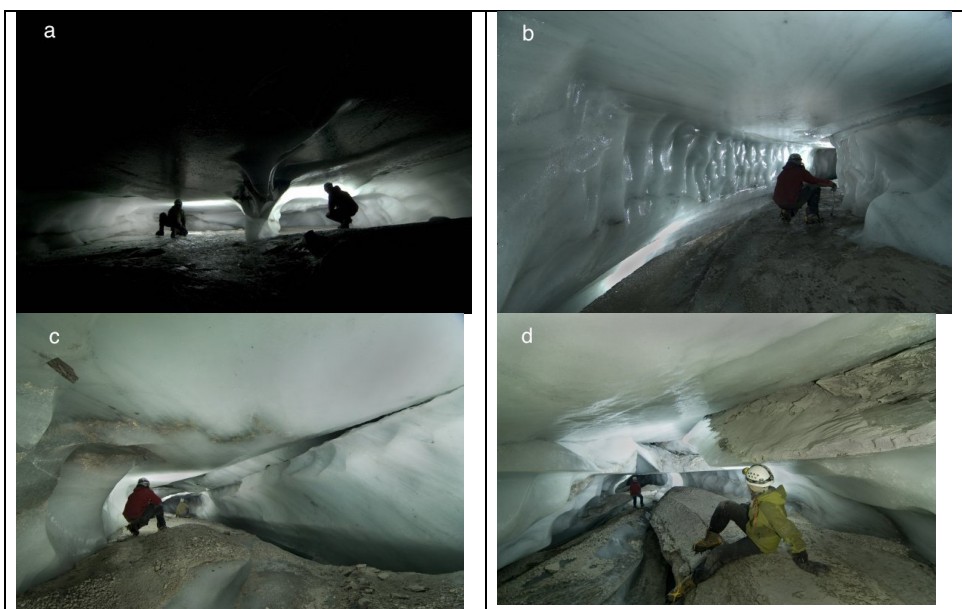

Fig. 4: Passage morphology in NG-04. a) Cutoff meander loop. Note inclined debris band on back wall behind the the left-hand figure. b) The upper passage near A12, showing suture between the right-hand wall and the ceiling, and the incised lower passage on the left. c) The upper passage near A7, with a void and suture between the right-hand wall and the ceiling. d) The upper passage near A6, showing a band of bedded sand filling a sub-horizontal suture above the foreground figure.

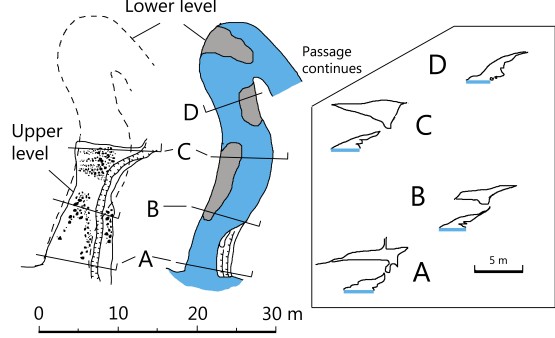

Fig. 5: Plan and passage cross-sections, conduit NG-05. 'Upper-' and Lower level' refer to the two floor levels indicated in cross-sections A-C. For location see Figure 12.





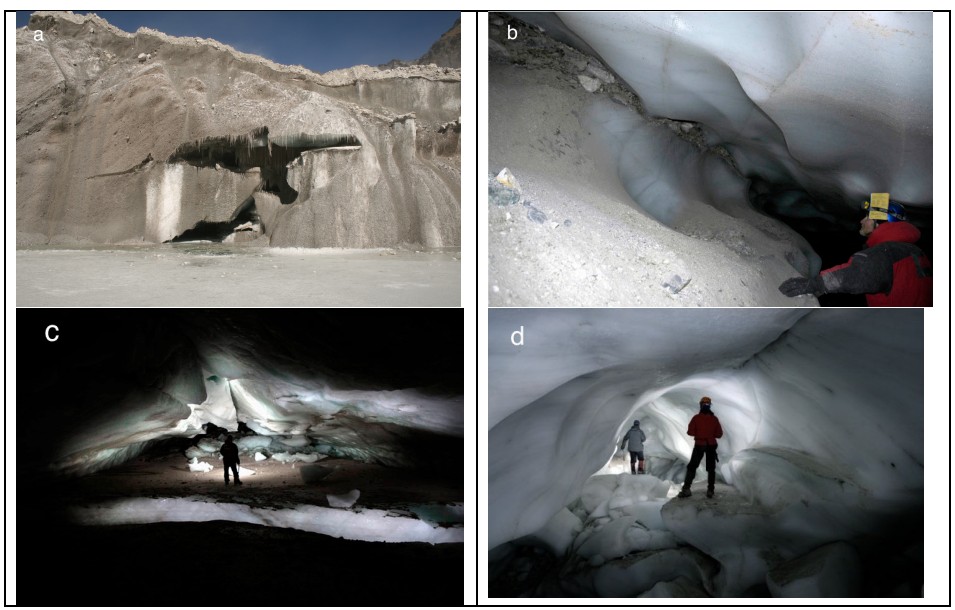

Fig. 6: a) The entrance of NG-05 on the NW margin of Spillway Lake; b) NG-01: debris-filled canyon suture at the upper level of the cave; c) NG-01: flat-floored mid level of the cave. Note canyon suture above and incised lower level crossing foreground from left to right; d) NG-02: Tubular upper passage with canyon suture in the roof.

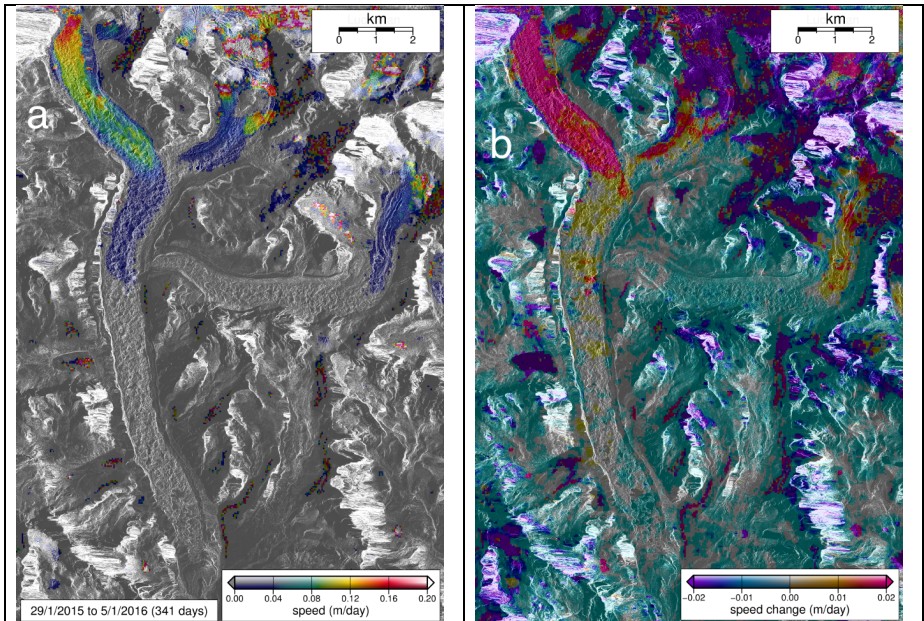

Fig. 7: Surface velocities derived from TerraSAR-X data: a) mean daily velocity for the 'annual' period 29 Jan 2015 to 5 Jan 2016; b) mean daily velocities for 29 Jan 2015 to 5 Jan 2016 minus mean daily velocities for 19 Sept 2014 to 18 Jan 2015, indicating minimum summer speed-up of the glacier.





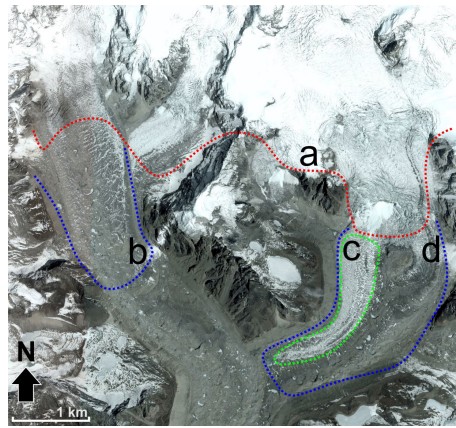

Fig. 8: The location of crevasse fields on the W and NE branches of Ngozumpa Glacier (a)
and areas where supraglacial channels occur on debris-covered (b, d) and clean (c) ice. Image
source: Google Earth

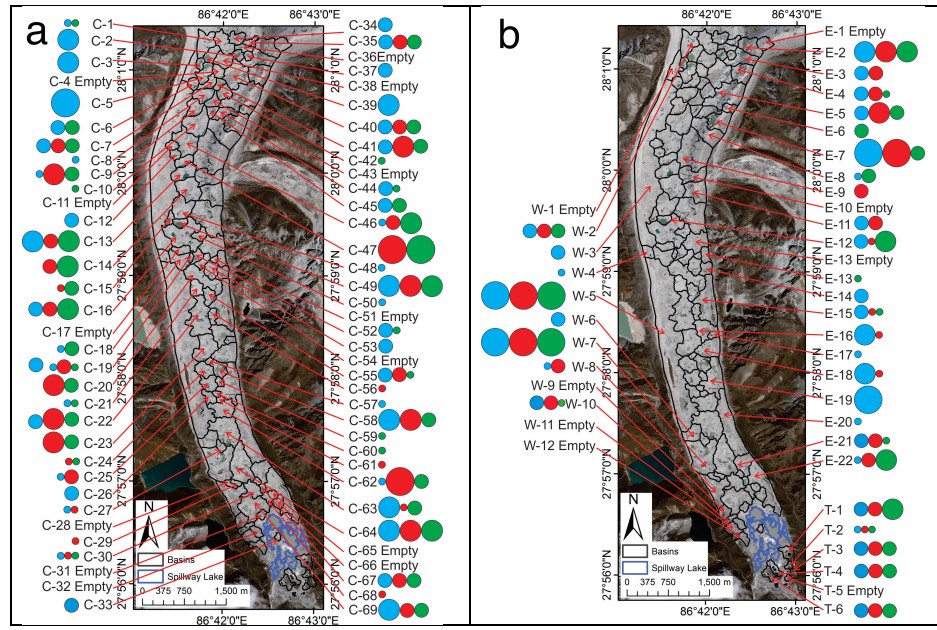

Fig. 9: Surface drainage basins and lake area changes: a) the central part of the glacier, and b)
the lateral margins and terminal zone. Lake areas are shown for 2010 (blue), 2012 (red) and
2015 (green), in four categories: <1000 m$^2$ (small circles), 1000-5000 m$^2$ (medium circles),
5000-10000 m$^2$ (large circles) and >10000 m$^2$ (largest circles).





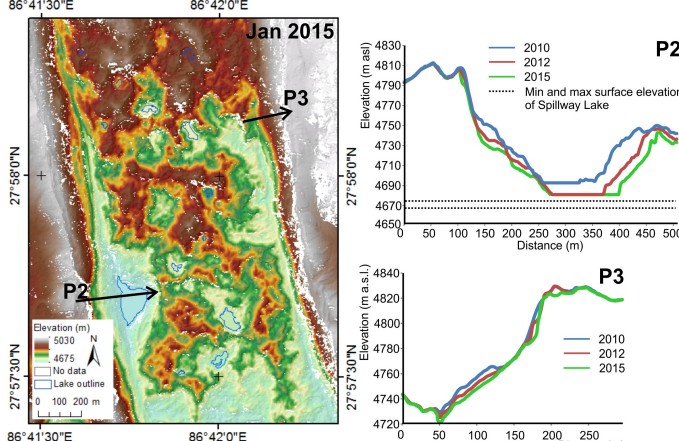

Fig. 10: Extract from the 2010 DEM and selected cross profiles in 2010, 2012 and 2015
showing lateral troughs, subsidence of trough floors and erosion of moraine slopes. Location
of the map is shown in Fig. 2.



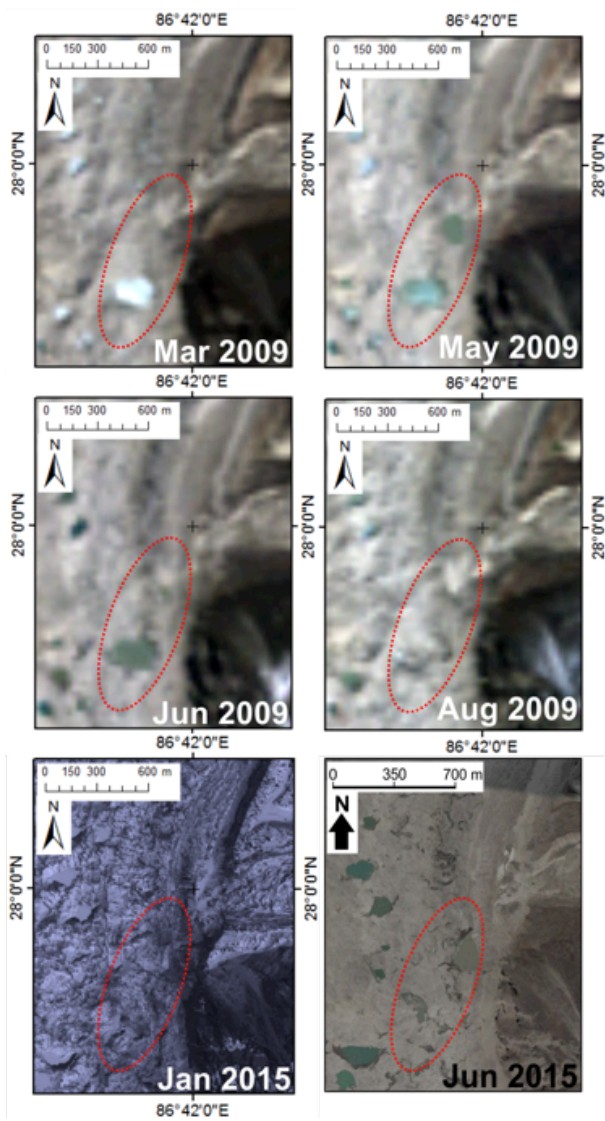

Fig. 11: Changing lake extent in Basin E-11, showing evidence of filling and drainage cycles,
on Landsat 5 TM (2009), WorldView-3 (Jan 2015) and (Jun 2015) imagery.



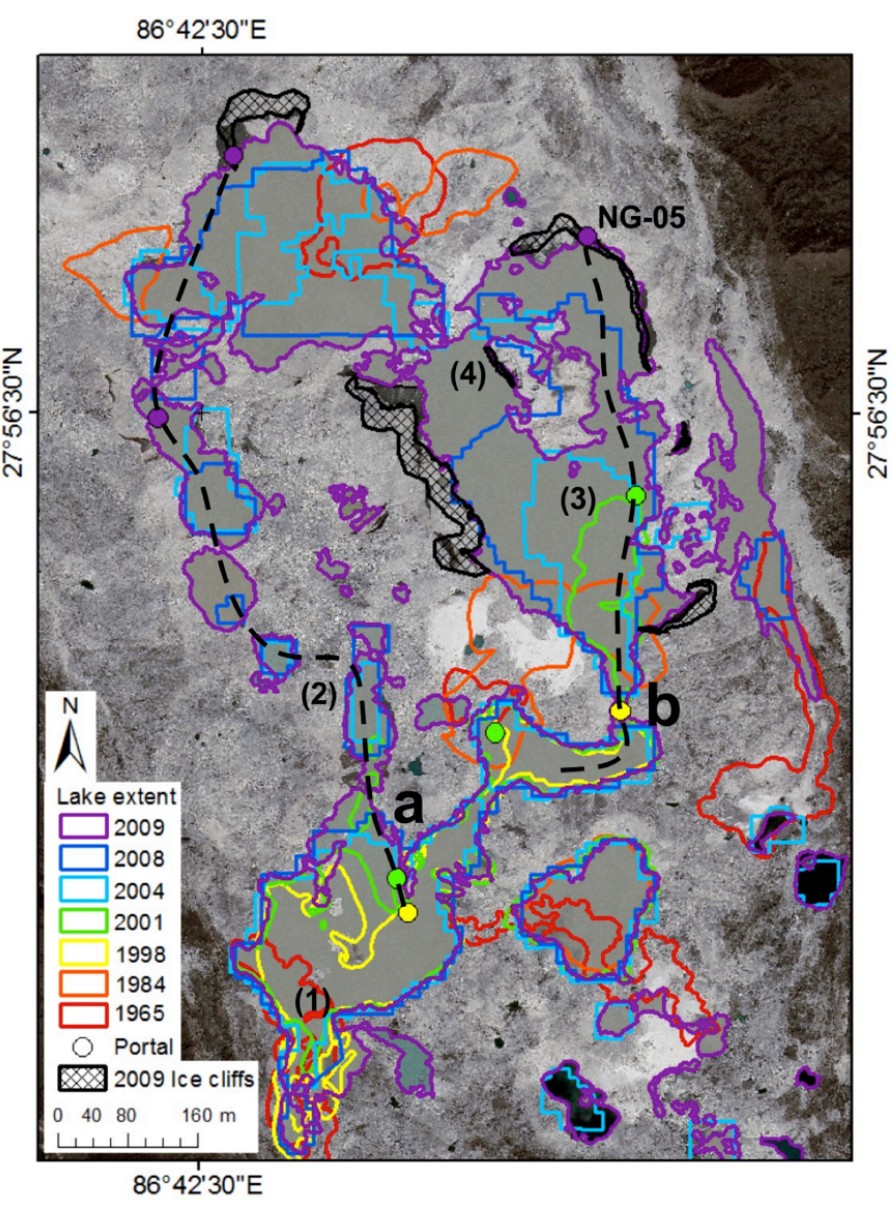

Fig. 12: Spillway Lake, 1965-2009, showing the position of meltwater portals and upwellings
and the inferred location of englacial conduits (dashed lines). Background image: GeoEye-1
from June 2010.





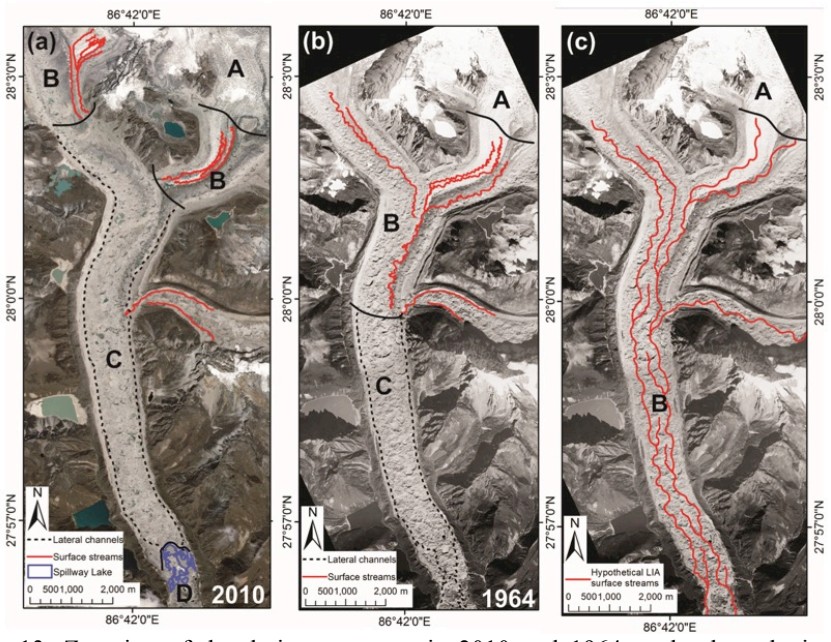

Fig. 13: Zonation of the drainage system in 2010 and 1964. and a hypothetical drainage system at the Little Ice Age maximum. A: crevasse fields; B: supraglacial channels; C: closed surface basins with perched lakes; D: Spillway lake. Dashed lines indicate the positions of sub-marginal conduits.

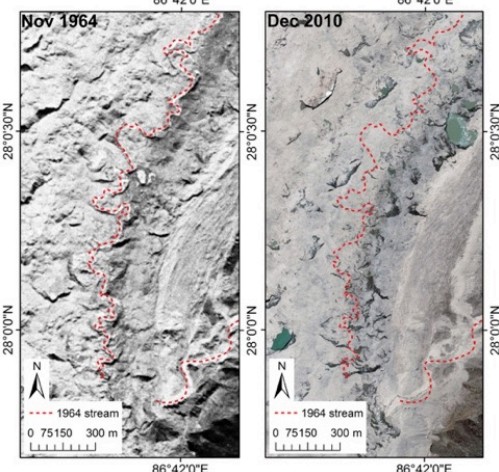

Fig. 14: Eastern margin of the main trunk upglacier from its confluence with the E branch, showing supraglacial channels (Corona imagery from 1964) and hummocky surface topography (GeoEye-1 imagery from 2010). For location of area see Fig. 2.