# Peer review of "Discussion started: 14 March 2017 © Author(s) 2017. CC-BY 3.0 License."

_The Cryosphere, 2017_

## Referee Comment (RC1) · Dr Sakai (Referee) · 11 Apr 2017

In this paper, authors have developed a comprehensive view on the drainage system of debris-covered glaciers in the Himalayas. Since almost part of drainage system are invisible by satellite images, they have linked various phenomena using geomorphic evidence (Google Earth, satellite images), and seasonal changes of surface velocity and speleological exploration. Although some parts of those links includes speculations, they attained a reasoned interpretation of view on the drainage system of debris-covered glaciers. I have never been to the Ngozumpa Glacier, but, the image was very understandable for me without difficulty. Therefore, I think the images of drainage system of debris-covered glaciers can be applied to other glaciers (as

authors wrote). I believe this paper will have great contribution to understand the drainage system of the debris-covered glacier (not only present but also transition process).

Please also note the supplement to this comment:
http://www.the-cryosphere-discuss.net/tc-2017-29/tc-2017-29-RC1-supplement.pdf

**Supplement:**

Comment on 'Structure and evolution of the drainage system of a Himalayan debris-covered glacier, and its relationship with patterns of mass loss' by Douglas I. Benn et al.

In this paper, authors have developed a comprehensive view on the drainage system of debris-covered glaciers in the Himalayas. Since almost part of drainage system are invisible by satellite images, they have linked various phenomena using geomorphic evidence (Google Earth, satellite images), and seasonal changes of surface velocity and speleological exploration. Although some parts of those links includes speculations, they attained a reasoned interpretation of view on the drainage system of debris-covered glaciers. I have never been to the Ngozumpa Glacier, but, the image was very understandable for me without difficulty. Therefore, I think the images of drainage system of debris-covered glaciers can be applied to other glaciers (as authors wrote). I believe this paper will have great contribution to understand the drainage system of the debris-covered glacier (not only present but also transition process).

<Major comment>
1) As you wrote 'Comparison of the drainage system structure in 2010 with evidence on Corona imagery from 1964 shows an upglacier expansion of the area occupied by closed depressions and perched lakes'(L623-625) I think analysis on the change of small basins (perched lake area) using Corona imagery (as past image) is useful to know the change of drainage system and to gain more insight of your synoptic view on the drainage system of debris-covered glaciers.
Iwata et al. (2000) have reported that high relief area expand from 1978 to 1995 at the middle ablation area of the Khumbu Glacier based on the geomorphic evidence. Although the target of Iwata's study was the Khumbu glacier (different from your target; the Ngozumpa Glacier), I think Iwata's result complement your result that lower limit of surface stream area has gone up to higher elevation recently. Therefore, Iwata et al. (2000) would be a nice reference of your manuscript.
http://iahs.info/uploads/dms/iahs_264_0003.pdf

2) In the section 4.2, 5.1 and 5.5, authors did not discuss on the start point (maximum altitude) of surface stream. The start point of surface stream strongly relates with altitude of ablation and it is significant for drainage system. I think discussion is required in the manuscript on the start point for example the difference between 1964 and 2010 (Fig. 13a and b).
Although there is no solid evidence on the surface stream during LIA, the start point of

surface stream during the LIA might (should?) be different from that of 1964. Please take into account the start point of surface stream in Fig. 13c.

<Specific comment>
L34-35 'since the 1970s'> Analysis period by Kääb et al., (2012) was 2003-2008.

L357-358 ' The seasonal variations in ice velocities in the upper ablation zone are too large to be explained by changes in ice creep rates,' > Authors should write the reason

L 375-385 It's better to cite Fig. 13 in this section.

L487 'measured volume losses' is ambiguous expression. Please write specifically. If the measured volume loss is calculated from elevation change at ablation area, the value includes not only ablation but also emergence velocity.

Fig 14 There is no symbols of a) and b) in the Fig. 14, although, authors used Fig. 14a, 14b in the text (L543-549)

L528 'where the overall gradient of the glacier is <3°' < reference?

L543 'By 2010, this part of the glacier had been broken up into basins E-7, E-8 and E-9'.
  > In other word, you can estimate that basins E-7, E-8 and E-9 has coalesced in 1960s from the Corona image. I recommend if you can draw the basins boundary using Corona image. it would be great help to understand the geomorphic change of the Ngozumpa Glacier. (main comment 1))

L594 'On the Survey of Nepal map,' > Reference is necessary, here. I think following map is cited here. 'Nepal: Survey Department. 1997c. Namuche Bajar 1 : 50 000. Kathmandu, Ministry of Land Reform and Management. Survey Department. (Sheet No. 2786 03.)' This map was produced based on the aerial photography taken in 1992.

L629 'Such a drainage system might have existed during the Little Ice Age, and persisted into the early 20th Century.' > I recommend that the supraglacial channels during the Little Ice Age is not based on Satellite imagery or other evidences. Therefore, the line of the supraglacial channels should be drawn by dotted lines.

<Reference>

Please check whole references in the text and in the reference list (not only following comment).

L39 Reynolds, 2000 > I could not find the reference in the reference list.

L146 Thompson et al. (2016) > I could not find the reference in the reference list.

L471 Horodyskuj (2015) > I could not find the reference in the reference list.

L730 The reference has no published year.( *Earth Science Reviews)*

L748 In the title of Gulley et al. 2009a, 'Mechanisms of ....' has been missed.

L800-805 There are two Quincey et al. (2005) but I could not find Quincey et al. (2005) in the body text.

---

## Referee Comment (RC2) · D.J. Quincey (Referee) · 18 Apr 2017

This paper brings together a range of observations, some of which have been previously published, in an attempt to describe the complete hydrological system of the Ngozumpa Glacier in Nepal. It formalises some existing ideas as well as proposing some new ones, which other people working in this region will be able to test and build on in their own work. Some of the interpretation is slightly speculative, and given the same datasets it may be that others would arrive at different conclusions, but this should not detract from the value of assembling them in a single analysis and effectively joining the dots between them, for the first time on a glacier such as this. It is very well-written and the figures are high-quality. It should definitely be published in my

view, but before it is there are a few gaps that could be closed up to make it an even stronger addition to the literature. These are detailed below, followed by some more minor comments and suggestions.

1. Part of the justification for the study (lines 63-71) is that we still know relatively little about englacial conduit formation, and specifically the relative importance of the three processes previously described in the two Gulley et al., 2009 papers and summarised in Benn et al., 2012. Although not stated explicitly, the subsequent analysis here suggests that cut-and-closure is the dominant, or even exclusive, mechanism, at least on Ngozumpa. Given that the argument against NG-01 to NG-03 being structurally controlled (lines 308-311) could be invoked for most conduits running parallel to flow, and that to my knowledge there have been no direct observations of hydrofracture here or in the wider region, some actual discussion of their relative importance would be an interesting addition to the manuscript. Based on their additional analysis, do the authors now believe that cut-and-closure is the dominant mechanism for these debris-covered glaciers, or does it just happen to prevail at Ngozumpa? Or is it paired, in that cut-and-closure forms the conduit in the first place, and then the relict channels provide the dominant structural control thereafter? Or some combination of these? Some clarification in the revised text would be a good addition.

2. The least-well constrained element of the paper is the analysis of the existence or otherwise of a subglacial hydrology, understandably so. In the absence of any direct observations, the velocity proxy provides some evidence for subglacial water in the upper ablation area, but if the authors are correct in their interpretation of this, what happens to it then? I'd be interested to see some further discussion of the lower ablation area – if as stated (line 507) all of the water leaving the glacier passes through Spillway Lake, do the authors propose that the subglacial waters from the upper ablation area pass through the lower ablation area and are then elevated at the terminus under pressure? Some direct measurement of the discharge would give an indication of whether it is at least the correct order of magnitude for a glacier of this size, but in

the absence of this some discussion of whether it might go to deep groundwater, or shallow groundwater and then emerge further down-glacier, would fill this gap. The hollow that drains the supraglacial channels (line 384) is also important in this regard – it hints at a direct surface-to-bed connection but there is no further information given – can any more light be shed on where these waters go?

3. The interpretation that hummocky closed basins cannot support a supraglacial hydrology is believable, but there is a spatial mismatch between the analysis shown in Figure 9 and the observations shown in Figure 8, which detracts from the argument. If this is the dominant control on supraglacial water (and the interpretation in Figure 13 hangs on it being so) then can the analysis in Figure 8 be extended up so we can see if it prevails in the upper ablation area too? Or at least see some morphological/topographic differences between c) and d) in Figure 8?

Minor comments

L86: 7922 m here but 7952 m in Figure 1.

L95: I suggest stating that it is 'effectively' stagnant, since it is probably deforming at some rate, just not detectable by the satellite analysis.

L130 and elsewhere: just a note that the terms pond and lake are used interchangeably – I'd have a preference for using the former for the majority, which are perched and relatively small, and saving the latter for Spillway Lake, which is not.

L158-167: can you add some more detail on the masking/filtering that is evident in Figure 7? And what is the threshold of 'detectable flow' referred to in line 95 – I guess that should be mentioned here too.

L331: this is 7 km earlier in the manuscript.

L351: should these artifacts not be masked as per Fig 7a? As it stands, it looks like you have greater confidence in the speed-up data than the annual pair measurements, which can't be right since you derive the former from the latter.

L364-366: there's some contradiction with the Figure caption here – the text says there are only crevasses in the upper part of the clean ice tongue (c), but the caption states they are only in (a)?

L371-373: this is the only hint at what happens to the subglacial water after it descends from the upper ablation area – can you offer any insight into where it may go then?

L382-385: this hollow is quite an important part of the picture, particularly if it shows the surface is connected to the bed directly – is it the same hollow that Horodyskuj monitored? Is it a moulin? Can you offer any further information on it?

L396: Figure 9 does not extend sufficiently far up-glacier for us to be able to verify this is true – can you extend the analysis to make this argument more robust?

L402-415: the number of basins etc is interesting here but the upper boundary of the analysis seems arbitrarily defined. Why not cover the whole of the debris-covered area? That way others can repeat the analysis for future time periods and quantify the change.

L410: Figure 9 doesn't show any full drainage events, unless these are lumped into the < 1000 m2 category? Shouldn't they be shown as 'empty'?

L425: do they have to be relict, necessarily?

L425-427: as above, I can't see the evidence for 35 drainage events – can you make this a bit clearer?

L424-427: can you add an acknowledgement that there is a seasonal signal in these data?

L439: they're probably underlain by thick sediment too, inhibiting bottom melt.

L456: this disparity between the number of basins on the west and the east sides merits some further comment – does it reflect the dominant englacial drainage pathway? Or debris-thickness? It's a stark contrast when looking at Figure 9.

L460: missing 'the'

L471: Horodyskuj (2015) is not in the reference list.

L507: can you be sure that all of the water passes through Spillway Lake?

L518-521: can you bring this into line with the six elements stated in the abstract? Or vice versa?

L528: have you derived this 3° threshold value earlier on? I can't find it in the previous analysis or discussion – can you qualify it somehow?

L543-544: it'd be better to show these basins superimposed on Fig 14b, rather than repeating the channels on both figure panels.

L545: can you indicate where these elongate ponds are on the figure for clarity?

L546-549: the figure referred to here doesn't relate to the text. Do you mean figure 10 instead?

L553: where does this thickening debris cover come from?

L557: note cut-and-closure is hyphenated in some places but not in others.

L589: section numbering jumps one here.

L626-627: remove the bracketed text since it is clear already?

L662-664: interesting that the Khumbu ponds also follow what might be a sub-marginal channel.

L666: this sentence implies that Ngozumpa is in a more advanced stage of recession than others in the region – is this what you mean? I'm not sure it is much different to others except for Spillway Lake?

L669: maybe 'interpretation' is better than 'view' here?

Figure 7: mentioned above too, but how can b) have more coverage than a) given that

it is derived from a)?

Figure 8: what are the coloured dashes here?

Figure 11: are there no better data than these TM images? Only because they're poor resolution. Do the 2010 and 2012 data you have not cover these areas? At least could you superimpose your interpreted pond boundaries?

Figure 12: maybe add that the reader should see the text for explanation of the annotations?

---

## Author Comment (AC1) · 5 Jun 2017

Many thanks for these constructive comments. We have revised the paper to take account of all of the points raised. Details of the changes are as follows:

1) As you wrote 'Comparison of the drainage system structure in 2010 with evidence on Corona imagery from 1964 shows an upglacier expansion of the area occupied by closed depressions and perched lakes'(L623-625) I think analysis on the change of small basins (perched lake area) using Corona imagery (as past image) is useful to know the change of drainage system and to gain more insight of your synoptic view on the drainage system of debris-covered glaciers.

[Figure]

Although it would be very interesting to analyse long-term basin evolution on the glacier, this is not possible. Our methods for defining basins use contoured DEMs, which cannot be constructed for the Corona data, which consist of mono images. Text has been added in L158-60 to explain this point.

Iwata et al. (2000) have reported that high relief area expand from 1978 to 1995 at the middle ablation area of the Khumbu Glacier based on the geomorphic evidence. Although the target of Iwata's study was the Khumbu glacier (different from your target; the Ngozumpa Glacier), I think Iwata's result complement your result that lower limit of surface stream area has gone up to higher elevation recently. Therefore, Iwata et al. (2000) would be a nice reference of your manuscript.

Reference to Iwata et al. (2000) has been added (L 687)

2) In the section 4.2, 5.1 and 5.5, authors did not discuss on the start point (maximum altitude) of surface stream. The start point of surface stream strongly relates with altitude of ablation and it is significant for drainage system. I think discussion is required in the manuscript on the start point for example the difference between 1964 and 2010 (Fig. 13a and b) there is no solid evidence on the surface stream during LIA, the start point of surface stream during the LIA might (should?) be different from that of 1964. Please take into account the start point of surface stream in Fig. 13c.

The upper limit of surface streams is not determined by the ELA, but the location of to-pographically controlled crevasse fields. The 1964 image shows these crevasse fields in the same location. Text has been added (L 651 ff.) to explain this point.

L34-35 'since the 1970s'> Analysis period by Kääb et al., (2012) was 2003-2008.

The position of Kääb reference has been changed to remove this problem.

L357-358 (365-8)' The seasonal variations in ice velocities in the upper ablation zone are too large to be explained by changes in ice creep rates,' > Authors should write the reason

justification of this statement has been added.

L 375-385 (387) It's better to cite Fig. 13 in this section.

A reference to Fig 14 (formerly 13) has been added.

L487 'measured volume losses' is ambiguous expression. Please write specifically. If the measured volume loss is calculated from elevation change at ablation area, the value includes not only ablation but also emergence velocity.

In the paper by Thompson et al. (2016) these volume losses were calculated from elevation changes on the stagnant part of the glacier, where complications from glacier flow do not arise. Rather than clutter the text with this information, we have added the phrase 'on the stagnant part of the glacier' on line 498.

Fig 14 There is no symbols of a) and b) in the Fig. 14, although, authors used Fig. 14a, 14b in the text (L543-549)

This Figure has been redrafted.

L528 'where the overall gradient of the glacier is <3°' < reference?

Statements about the glacier gradient have been clarified (L 106), and a new Figure 3 added.

L543 'By 2010, this part of the glacier had been broken up into basins E-7, E-8 and E-9'. > In other word, you can estimate that basins E-7, E-8 and E-9 has coalesced in 1960s from the Corona image. I recommend if you can draw the basins boundary using Corona image. it would be great help to understand the geomorphic change of the Ngozumpa Glacier. (main comment 1))

This Figure has been redrafted to make the changes clearer.

L594 'On the Survey of Nepal map,' > Reference is necessary, here. I think following map is cited here. 'Nepal: Survey Department. 1997c. Namuche Bajar 1 : 50 000.

Kathmandu, Ministry of Land Reform and Management. Survey Department. (Sheet No. 2786 03.)' This map was produced based on the aerial photography taken in 1992.

Reference added.

L629 'Such a drainage system might have existed during the Little Ice Age, and persisted into the early 20th Century.' > I recommend that the supraglacial channels during the Little Ice Age is not based on Satellite imagery or other evidences. Therefore, the line of the supraglacial channels should be drawn by dotted lines.

This has been done.

Please check whole references in the text and in the reference list (not only following comment).

L39 Reynolds, 2000 > I could not find the reference in the reference list.

Reference deleted in text.

L146 Thompson et al. (2016) > I could not find the reference in the reference list.

Reference added.

L471 Horodyskuj (2015) > I could not find the reference in the reference list.

Reference added

L730 The reference has no published year.( Earth Science Reviews)

Year of publication added.

L748 In the title of Gulley et al. 2009a, 'Mechanisms of ....' has been missed.

Text corrected

L800-805 There are two Quincey et al. (2005) but I could not find Quincey et al. (2005) in the body text.

[Figure]

This has been reduced to one reference, with the correct year of 2007

---

## Author Comment (AC2) · 5 Jun 2017

Thank you for these detailed and constructive comments. We have revised the text and Figures to take the comments into account, as detailed below.

1. Part of the justification for the study (lines 63-71) is that we still know relatively little about englacial conduit formation, and specifically the relative importance of the three processes previously described in the two Gulley et al., 2009 papers and summarised in Benn et al., 2012. Although not stated explicitly, the subsequent analysis here suggests that cut-and-closure is the dominant, or even exclusive, mechanism, at least on Ngozumpa. Given that the argument against NG-01 to NG-03 being structurally controlled (lines 308-311) could be invoked for most conduits running parallel to flow, and that to my knowledge there have been no direct observations of hydrofracture here or in the wider region, some actual discussion of their relative importance would be an interesting addition to the manuscript. Based on their additional analysis, do the authors now believe that cut-and-closure is the dominant mechanism for these debris-covered glaciers, or does it just happen to prevail at Ngozumpa? Or is it paired, in that cut-and-closure forms the conduit in the first place, and then the relict channels provide the dominant structural control thereafter? Or some combination of these? Some clarification in the revised text would be a good addition.

A paragraph has been added (L 610-17) to clarify these points.

2. The least-well constrained element of the paper is the analysis of the existence or otherwise of a subglacial hydrology, understandably so. In the absence of any direct observations, the velocity proxy provides some evidence for subglacial water in the upper ablation area, but if the authors are correct in their interpretation of this, what happens to it then? I'd be interested to see some further discussion of the lower ablation area – if as stated (line 507) all of the water leaving the glacier passes through Spillway Lake, do the authors propose that the subglacial waters from the upper ablation area pass through the lower ablation area and are then elevated at the terminus under pressure? Some direct measurement of the discharge would give an indication of whether it is at least the correct order of magnitude for a glacier of this size, but in the absence of this some discussion of whether it might go to deep groundwater, or shallow groundwater and then emerge further down-glacier, would fill this gap. The hollow that drains the supraglacial channels (line 384) is also important in this regard – it hints at a direct surface-to-bed connection but there is no further information given – can any more light be shed on where these waters go?

Although we lack sufficient data to answer these questions, we have added speculations (L381 & 505-8) to address the issue of water routing.

3. The interpretation that hummocky closed basins cannot support a supraglacial hydrology is believable, but there is a spatial mismatch between the analysis shown in Figure 9 and the observations shown in Figure 8, which detracts from the argument. If this is the dominant control on supraglacial water (and the interpretation in Figure 13 hangs on it being so) then can the analysis in Figure 8 be extended up so we can see if it prevails in the upper ablation area too? Or at least see some morphological/topographic differences between c) and d) in Figure 8.

There is indeed a gap between areas where channels are visible on the glacier surface and area where surface basins can be reliably mapped. We have addressed this mismatch by identifying a 'transitional zone', now discussed in the text (L 411 ff.), and shown on Fig. 9 (previously Fig. 8) and a new Figure 3.

L86: 7922 m here but 7952 m in Figure 1.

>The true elevation is 7952 m, and this has been corrected in the text.

L95: I suggest stating that it is 'effectively' stagnant, since it is probably deforming at some rate, just not detectable by the satellite analysis.

>The change has been made

L130 and elsewhere: just a note that the terms pond and lake are used interchangeably – I'd have a preference for using the former for the majority, which are perched and relatively small, and saving the latter for Spillway Lake, which is not.

> Done as suggested.

L158-167: can you add some more detail on the masking/filtering that is evident in Figure 7? And what is the threshold of 'detectable flow' referred to in line 95 – I guess that should be mentioned here too.

> No masking or filtering was applied. A note on the definition of 'detectable motion' has been added in L171.
L331: Length of stagnant zone - this is 7 km earlier in the manuscript.

> This has been changed to 6.5 km in both cases.

L351: should these artifacts not be masked as per Fig 7a? As it stands, it looks like you have greater confidence in the speed-up data than the annual pair measurements, which can't be right since you derive the former from the latter.

> Neither the absolute nor differenced velocity maps have been filtered, and the existence of velocity difference data in areas of no apparent absolute annual displacement is simply a result of the scales used. A note has been added to the caption of Fig. 7 to point out the lack of masking and filtering.

L364-366: there's some contradiction with the Figure caption here – the text says there are only crevasses in the upper part of the clean ice tongue (c), but the caption states they are only in (a)?

> The text has been corrected.

L371-373: this is the only hint at what happens to the subglacial water after it descends from the upper ablation area – can you offer any insight into where it may go then? Lines added to 375, and 491 ff, to speculate on fate of subglacial water.

> We have added the speculation that the subglacial water is drained via the submarginal channels.

L382-385: this hollow is quite an important part of the picture, particularly if it shows the surface is connected to the bed directly – is it the same hollow that Horodyskuj monitored? Is it a moulin? Can you offer any further information on it?

> This is not the same hollow, as is now made clear in the text. We have no evidence that direct surface to bed drainage occurs in any of these basins, and decline to speculate about this issue.

L396: Figure 9 does not extend sufficiently far up-glacier for us to be able to verify this

is true – can you extend the analysis to make this argument more robust?

> Basins cannot be reliably delineated any further upglacier. Text and a new figure have been added to explain this in terms of a 'transitional zone'. (see reply to Point 3 above)

L402-415: the number of basins etc is interesting here but the upper boundary of the analysis seems arbitrarily defined. Why not cover the whole of the debris-covered area? That way others can repeat the analysis for future time periods and quantify the change.

> see previous comment

L410: Figure 9 doesn't show any full drainage events, unless these are lumped into the < 1000 m2 category? Shouldn't they be shown as 'empty'?

> Empty basins in any one year are indicated by 'missing' coloured circles - The caption has been modified to explain this.

L425: do they have to be relict, necessarily?

> 'relict' has been changed to 'active or relect'

L425-427: as above, I can't see the evidence for 35 drainage events – can you make this a bit clearer?

> see comment on line 410.

L424-427: can you add an acknowledgement that there is a seasonal signal in these data?

> We do not have a good enough data series to reliably identify a seasonal signal.

L439: they're probably underlain by thick sediment too, inhibiting bottom melt.

> We do not have any data on pond bottom sediment thickness, so decline to comment. L456: this disparity between the number of basins on the west and the east sides merits

some further comment – does it reflect the dominant englacial drainage pathway? Or debris-thickness? It's a stark contrast when looking at Figure 9.

> This is indeed striking, but the reason is unknown. We think that speculation neither justified nor helpful.

L460: missing 'the'

> added

L471: Horodyskuj (2015) is not in the reference list.

> Added

L507: can you be sure that all of the water passes through Spillway Lake?

> A sentence has been added to qualify this statement.

L518-521: can you bring this into line with the six elements stated in the abstract?

> Done.

L528: 3° gradient.

> A statement about glacier gradients (line 105) and a new Figure 3 has been added to clarify this issue.

L543-544: it'd be better to show these basins superimposed on Fig 14b, rather than repeating the channels on both figure panels.

> The figure has been redrafted to make the evolution of this part of the glacier clearer. L545: can you indicate where these elongate ponds are on the figure for clarity?

Done.

L546-549: the figure referred to here doesn't relate to the text. Do you mean figure 10 instead? clarify

[Figure]

This has been corrected.

L553: where does this thickening debris cover come from?

> Text has been added to explain this.

L557: note cut-and-closure is hyphenated in some places but not in others.

> Changed to 'cut-and-closure' throughout

L589: section numbering jumps one here.

> The numbering has been corrected.

L626-627: remove the bracketed text since it is clear already?

> Bracketed text has been removed.

L662-664: interesting that the Khumbu ponds also follow what might be a sub-marginal channel.

Agreed. This is something for others to investigate in detail.

L666: this sentence implies that Ngozumpa is in a more advanced stage of recession than others in the region – is this what you mean? I'm not sure it is much different to others except for Spillway Lake?

> Not all others - for example, Imja and Trakarding have large base-level lakes. Base level lakes do not yet exist on Khumbu Glacier. Spillway Lake is an important example of a transitional stage.

L669: maybe 'interpretation' is better than 'view' here?

> Changed as requested.

Figure 7 (now 8): mentioned above too, but how can b) have more coverage than a) given that it is derived from a)?

> The difference map also depends on the winter velocities (not shown). Also, the velocity scale in a) fades to grey, so small velocities (which can contribute to velocity differences) are not visible. Text has been added (L 358) to point out that the small changes indicated on the lower glacier are below the margin of error.

Figure 8: what are the coloured dashes here?

> They delineate the areas identified by the letters.

Figure 11 (now 12): are there no better data than these TM images? Only because they're poor resolution. Do the 2010 and 2012 data you have not cover these areas? At least could you superimpose your interpreted pond boundaries?

> Pond boundaries superimposed to make the figure clearer.

Figure 12 (now 13): maybe add that the reader should see the text for explanation of the annotations?

> Text added to caption

---

## Author Comment (AC3) · 5 Jun 2017

The comment was uploaded in the form of a supplement:
http://www.the-cryosphere-discuss.net/tc-2017-29/tc-2017-29-AC3-supplement.pdf

---

## Author Response (AR2)

Dear Editor,

I have now made all of the technical very minor comments/corrections you requested, detailed as follows. Because these are minor, I have not included a marked up version of the paper in this reply.

Changes:

p. 3. Line 57: typo, delete the '§' after 'those'

done

p. 5: citation of figure 3 (line 106) is now first and before figure 2 (line 108), so maybe swap the order of the two figures.

done, and all references to these figures changed accordingly

p. 17 line 425: 'GREAT stability' is a bit awkward, maybe '...clearly exhibited stability' is better.

done

p. 21 line 526: the full stop should be AFTER the bracket instead of before! E.g. '...ramp).'

The text is correct as it stands. Full stops should be inside brackets if the brackets contain a complete sentence, outside if the brackets contain part of a sentence.

p. 24 line 610: this sentence ((cut-closure dominant process) probably only applies for 'debris covered area', if so should be made clear here.

wording changed to make this point clear

in acknowledgments: maybe acknowledge input from referees.

done

Fig. 1: improve resolution/quality of Figure in final submission, any lines/writing are pretty blurred and the writing in the inset figure is almost unreadable.

The full resolution figure file is included in the submission

Fig. 4: clarify in caption the difference between grey filled and white filled (white dashed lines). I guess they refer to upper versus lower level width of the channel. Further, first line in caption (line 917): change text to '...sections of englacial conduit...' (rather than '...sections. englacial conduit...').

Text has been added to the caption to explain all symbols.

Fig. 6: the solid outlined channel refers to upper level outlines but then the I assume the drawn boulders are probably lying on floor. So with upper level it means still the floor of the channel and not the ceiling?

Text added for clarification.

Fig. 8 caption (line 937-938): I would rather say '...b) velocity difference between 'annual' period (29 Jan. 2015 to 5 Jan 2016') and 'winter' period (19 Sept 2014 and 18 Jan 2015), indicating...'

done

Fig. 9: again improve resolution/quality of figure in final submission, right now I can not see any crevasses (only guess them). Further, crevasses probably occur AROUND the red line (not just AT the redline). the redline does probably refer

text has been modified to clarify what the image shows, and the meaning of the red line.

Fig. 11: again, improve resolution/quality of Figure in final submission, any lines/writing are pretty blurred and the labels hard to read.

The full resolution figure file is included in the submission

with best wishes,

Doug Benn